# Activated Carbons from Hydrothermal Carbonization and Chemical Activation of Olive Stones: Application in Sulfamethoxazole Adsorption

Elena Diaz [1,*], Ines Sanchis [1], Charles J. Coronella [2] and Angel F. Mohedano [1]

[1] Chemical Engineering Department, Faculty of Sciences, Universidad Autonoma de Madrid, 28049 Madrid, Spain; ines.sanchis@uam.es (I.S.); angelf.mohedano@uam.es (A.F.M.)

[2] Department of Chemical and Materials Engineering, University of Nevada, Reno, NV 89557, USA; coronella@unr.edu

* Correspondence: elena.diaz@uam.es

**Abstract:** This work focuses on the production of activated carbons by hydrothermal carbonization of olive stones at 220 °C, followed by chemical activation with KOH, FeCl$_3$ and H$_3$PO$_4$ of the hydrochar obtained. In addition, N-doped hydrochars were also obtained by performing the hydrothermal carbonization process with the addition of $(NH_4)_2SO_4$. All hydrochars, N-doped and non-doped, showed low BET surface areas (4–18 m$^2$ g$^{-1}$). Activated hydrochars prepared using H$_3$PO$_4$ or KOH as activating agents presented BET surface areas of 1115 and 2122 m$^2$ g$^{-1}$, respectively, and those prepared from N-doped hydrochar showed BET surface area values between 1116 and 2048 m$^2$ g$^{-1}$ with an important contribution of mesoporosity (0.55–1.24 cm$^3$ g$^{-1}$). The preparation procedure also derived inactivated hydrochars with predominantly acidic or basic groups on their surface. The resulting materials were tested in the adsorption of sulfamethoxazole in water. The adsorption capacity depended on both the porous texture and the electrostatic interactions between the adsorbent and the adsorbate. The adsorption equilibrium data (20 °C) fitted fairly well to the Langmuir equation, and even better to the Freundlich equation, resulting in the non-doped hydrochar activated with the KOH as the best adsorbent.

**Keywords:** activated carbon; adsorption; hydrochar; hydrothermal carbonization; N-doped materials; olive stones; sulfamethoxazole

## 1. Introduction

Hydrothermal carbonization (HTC) is becoming an increasingly attractive way for the valorization of wet biomass wastes, which is carried out in the presence of water, at temperatures between 180 and 250 °C and the corresponding saturation pressure, and residence times between 5 min to 24 h [1–3]. The main reaction product is a solid known as hydrochar (HC), which is more stable and has a higher carbon content than the raw biomass. In addition, HTC produces a high organic loading of process water and a minimal gas fraction consisting mainly of CO$_2$. Hydrochar is commonly used: (i) for soil amendment [4]; (ii) as a source of energy [5–7]; or (iii) as a precursor of activated carbon materials [8–10]. In this sense, hydrothermal carbonization can be considered as a biomass pre-processing treatment to alter precursor properties prior to activation. HTC maintains the surface chemical functionality in the hydrochar and allows a more complete activation by favoring the development of a high porosity in the activation stage [11,12]. Physical activation consists of a high-temperature treatment (800–1100 °C) in a partially oxidizing atmosphere using steam, air or CO$_2$. Chemical activation requires the use of chemical reagents (KOH, ZnCl$_2$, K$_2$CO$_3$, H$_3$PO$_4$), a subsequent thermal treatment (usually 500–850 °C) for 1–24 h in an inert atmosphere and a final washing (usually with acid and water) to remove the excess activating agent [12–14]. As a result of the activation process, an increase in the

specific surface area of carbonaceous materials is achieved, which is more significant for chemical activation, and structural modifications and changes in their composition are also observed, such as the loss of oxygen functional groups or an increase in ash content [15].

Recently, N doping of hydrochar during the hydrothermal process has been used to modify the physical and chemical properties of hydrochar (e.g., increase in aromatic character, bulk N content, oxidation resistance and conductivity), without causing significant changes in the textural properties of hydrochar. Doped hydrochar requires an activation process to increase pore volume and surface area [16–20]. The incorporation of nitrogen into carbon structures has been shown to influence the performance of carbon materials used as energy-storage materials [21,22] and adsorbents [20,23–26]. Roldan et al. [20] prepared N-doped activated carbon by hydrothermal carbonization with $ZnCl_2$ as an activating agent (190 °C, 19 h) using glucose as a C precursor and pyrrole carboxaldehyde ($C_5H_5NO$) as a doping agent. They observed that the N-doped material showed higher mesoporosity (BET surface area ($A_{BET}$) = 503 $m^2$ $g^{-1}$, $V_{mesopore}$ = 0.44 $cm^3$ $g^{-1}$, pore diameter = 12 nm, N = 3.9 wt.%) than the non-modified hydrochar ($A_{BET}$ = 373 $m^2$ $g^{-1}$, $V_{mesopore}$ = 0.14 $cm^3$ $g^{-1}$, pore diameter = 3.5 nm, N = 0 wt.%). This property was responsible for the higher adsorption capacity ($q_e$) of N-doped carbon ($q_e$ = 159 mg $g^{-1}$ methylene blue; $q_e$ = 105 mg $g^{-1}$ Rhodamine B) than the non-modified material ($q_e$ = 88 mg $g^{-1}$ methylene blue; $q_e$ = 13 mg $g^{-1}$ Rhodamine B). The porosity and N content of N-doped porous carbons can be tuned by adjusting the doping agent and C precursor ratio. In this regard, Huang et al. [26] prepared an N-doped carbon using $NaNH_2$ as a doping agent and a hydrochar from furfural. They doped the material with a $NaNH_2$/hydrochar mass ratio of 2, 3 and 4 at 600 °C. The results showed an increase in porosity from 1068 to 2436 $m^2$ $g^{-1}$ with the ratio 2 (N = 0.78 wt.%) and 4 (N = 4.30 wt.%), achieving a $CO_2$ adsorption capacity (100 kPa and 0 °C) of 4.5 and 5.4 mmol $g^{-1}$, respectively.

For the preparation of low-cost adsorbents from biomass waste, both the amount of waste generated, and its location must be considered. Spain is the world's leading source of olives. The olive industry is an area of intensive agricultural activity and there are by-products that are not widely used and accumulate as waste. In the 2020–2021 campaign, 2.75 million hectares have been dedicated to achieving an olive production of 1.39 million tons [27]. Considering that the weight of the olive stone represents between 10–20 % of the total weight of the olive, the generation in this period of olive stones could exceed 200,000 tons. Olive stones are being widely used as fuel in southern Europe, especially in Croatia, Greece, Italy, Portugal, Slovenia, Spain and Turkey [28,29]. Recently, different alternatives have emerged to give value to this by-product as the production of catalysts [30], food supplements [31] and adsorbents for $CO_2$ removal [32,33], heavy metals [34,35], textile dyes [36] and pharmaceuticals [37,38]. Several preparation methods have been carried out to transform olive stones into a good adsorbent material. Moussa et al. [32,33] prepared activated carbon from olive stones by a first pyrolysis step (300 °C, 1 h, $N_2$) of the raw material, followed by KOH impregnation (85 °C, 3 h, KOH/OS mass ratio = 7) and a second pyrolysis step (350 °C, 2 h, $N_2$). They obtained activated carbons with an $A_{BET}$ of 1345 $m^2$ $g^{-1}$, with an adsorption capacity of 5.7 mmol $g^{-1}$ for $CO_2$ (0 °C, 1 bar), higher than those of other activated carbons prepared from algae ($A_{BET}$ = 418 $m^2$ $g^{-1}$, $q_e$ = 2.4 mmol $g^{-1}$) [39] or from empty fruit bunches of oil palms ($A_{BET}$ = 2510 $m^2$ $g^{-1}$, $q_e$ = 5.2 mmol $g^{-1}$) [40]. Bohli et al. [34] impregnated olive stones with $H_3PO_4$ (110 °C, 9 h) followed by a calcination process (380 °C, 2.5 h, $N_2$). They obtained an activated carbon characterized by an $A_{BET}$ of 1194 $m^2$ $g^{-1}$ and a $pH_{PZC}$ of 3.4, which was a good adsorbent to remove Cu ($q_e$ = 17.7 mg $g^{-1}$), Cd ($q_e$ = 57.1 mg $g^{-1}$) and Pb ($q_e$ = 147.5 mg $g^{-1}$). These results were appreciably better than those obtained by other authors with apricot stones ($A_{BET}$ = 566 $m^2$ $g^{-1}$, $q_e$ = 24.1, 33.6 and 22.8 mg $g^{-1}$ for Cu, Cd and Pb, respectively) [41]. The same preparation process was used by Limousy et al. [38] to prepare activated carbons for amoxicillin adsorption, obtaining a material ($A_{BET}$ = 1174 $m^2$ $g^{-1}$) able to remove 93% of the antibiotic (20 °C for 25 mg $L^{-1}$ initial concentration) with an adsorption capacity of 22.1 mg $g^{-1}$.

The aim of this work is the valorization of olive stones into activated carbon by means of hydrothermal carbonization and chemical activation of the resulting hydrochar. The effect of the N doping of the carbonaceous materials along the hydrothermal carbonization process on the characteristics of the adsorbent materials has been also studied. Hydrochars and activated hydrochars have been characterized by several techniques covering proximate and ultimate analyses, $N_2$ adsorption-desorption at 77 K, Scanning Electron Microscope (SEM) and pH$_{slurry}$. The potential of the activated hydrochars as adsorbents for the removal of pollutants in water has been tested using as a model compound an emerging pollutant, the sulfamethoxazole (SMX), a sulfonamide antibiotic used to treat urinary tract infections, bronchitis and prostatitis, which is effective against gram-negative and positive bacteria. That compound is frequently detected in wastewater and in drinking water at concentrations of 100–2500 ng L$^{-1}$ and 12 ng L$^{-1}$, respectively [42]. Table 1 collects some SMX adsorption results using non-commercial activated carbons from biomass wastes as adsorbents. The reported materials showed low specific surfaces, except those that follow preparation procedures based on the hydrothermal carbonization of biomass wastes and KOH chemical activation or pyrolysis at high temperatures. The results suggest that the electrostatic interaction between functionalized adsorbents and SMX plays an important role in pollutant removal.

**Table 1.** Summary of representative studies on sulfamethoxazole adsorption from water by biomass-derived adsorbents (the data correspond to the adsorbent that exhibits the best performance in each study).

| Biomass Precursor | Adsorbent Preparation | Adsorbent Characteristics | Adsorption Conditions/Parameters | Ref. |
|---|---|---|---|---|
| Bagasse | Magnetic biochar prepared by $FeCl_3$ impregnation and pyrolysis at 800 °C | C = 73.8% O = 21.8% N = 0.96% pH$_{PZC}$ = 2.8 A$_{BET}$ = 606 m$^2$ g$^{-1}$ | SMX$_0$ = 50 mg L$^{-1}$ W = 0.2 g L$^{-1}$ T = 25 °C pH = 5 $q_L$ = 205 mg g$^{-1}$ | [43] |
| Dewatered waste activated sludge | Hydrothermal carbonization at 208 °C and KOH activation at 850 °C | C = 34.9% N = 5.8% pH$_{slurry}$ = 5.5 A$_{BET}$ = 832 m$^2$ g$^{-1}$ | SMX$_0$ = 25–175 mg L$^{-1}$ W = 0.25 g L$^{-1}$ T = 20 °C pH = 4.6 $q_L$ = 423 mg g$^{-1}$ | [44] |
| Grape Seeds | Hydrothermal carbonization at 220 °C and KOH activation at 750 °C | C = 73.3% pH$_{slurry}$ = 7.6 A$_{BET}$ = 2194 m$^2$ g$^{-1}$ | SMX$_0$ = 25–150 mg L$^{-1}$ W = 0.25 g L$^{-1}$ T = 20 °C pH = 4.6 $q_L$ = 650 mg g$^{-1}$ | [45] |
| Pine sawdust | Magnetic biochar prepared by $FeCl_2$, KOH and $KNO_3$ impregnation at 90 °C | C = 55.8% O = 14.2% N < 0.3% pH$_{PZC}$ = 9.5 A$_{BET}$ = 126 m$^2$ g$^{-1}$ | SMX$_0$ = 0.5–9.0 mg L$^{-1}$ W = 20 mg L$^{-1}$ T = 25 °C pH = 4.5 $q_L$ = 19.1 mg g$^{-1}$ | [46] |
| Bamboo | Thermal treatment ($N_2$) at 380 °C and 2.5–10 psi followed by an activation with $H_3PO_4$ at 600 °C | C = 52.0% O = 39.5% A$_{BET}$ = 1.12 m$^2$ g$^{-1}$ | SMX$_0$ = 10 mg L$^{-1}$ W = 100 mg L$^{-1}$ T = 25 °C pH = 3.3 $q_L$ = 88.1 mg g$^{-1}$ | [47] |
| Pine wood | Carbonization at 500 °C | C = 87.6% O = 12.2% A$_{BET}$ = 328 m$^2$ g$^{-1}$ | W = 0.45–0.68 mg L$^{-1}$ Room temperature pH = 6 $K_F$ = 131 $n$ = 0.24 | [48] |

A$_{BET}$: specific surface area by Brunauer–Emmett–Teller (BET) equation; C: carbon content (wt.% d.b.); O: oxygen content (wt.% d.b.); N: nitrogen content (wt.% d.b.); Ash: ash content (wt.% d.b.); pH$_{PZC}$: pH at point of zero charge; SMX$_0$: initial sulfamethoxazole concentration; W: adsorbent dose; T: adsorption temperature; pH: adsorption pH; $q_L$; adsorption capacity from Langmuir model; $K_F$: Freundlich constant; $n$: adsorption intensity from Freundlich model.

## 2. Materials and Methods

### 2.1. Preparation of Hydrochars

Olive stones (OS), provided by a company located in Jaén (Spain), with an 18 wt.% of humidity and with a particle size of 5 mm were used as hydrochar precursors. In the first stage, the HTC process was carried out in a Teflon-lined stainless steel vessel (100 mL), using 20 g of dried olive stones and multiple water amounts to obtain an OS concentration in the reactor between 30–50 wt.% on a dry basis. The reactor was inserted in a muffle furnace (Hobersal series 8B Mod 12 PR/400) and heated up to 220 °C for 16 h [45]. The hydrochar was recovered by filtration, washed with distilled water and oven-dried at 105 °C for 24 h (Nabertherm R 60/750/12-C6).

Nitrogen doping of hydrochars was carried out using 20 g of dried OS and 50 wt.% water spiked with 7.9, 13.2 and 19.8 g of $(NH_4)_2SO_4$ (Panreac, 99%) in an initial C/N mass ratio of 5, 3 and 2, respectively [19]. HTC process and subsequent hydrochar washing were carried out under the same conditions abovementioned. Hydrochars were denoted as HC, followed by NX (HCNX) in cases of N-doped hydrochars, where X represents the initial C/N mass ratio.

### 2.2. Preparation of Activated Hydrochars

The hydrochar was physically mixed with the different activating agents, such as potassium hydroxide (KOH, Panreac, 85%), iron chloride ($FeCl_3$, Panreac, 97%) or phosphoric acid ($H_3PO_4$, Panreac, 85%) in a 3:1 mass ratio at room temperature. They were activated in a horizontal tube furnace (Nabertherm RHTH 120/300/18/C42) under a continuous $N_2$ flow (100 NmL min$^{-1}$). Activation with KOH and $FeCl_3$ was performed at 750 °C for 1 h with a heating rate of 10 °C min$^{-1}$ [13,49,50]. In the case of $H_3PO_4$, the mixture was left overnight at 60 °C and then heated at 500 °C (10 °C min$^{-1}$ heating rate) for 2 h [51]. In addition, chemical activation with KOH, $FeCl_3$ and $H_3PO_4$ of the non-carbonized olive stones was carried out following the abovementioned procedure. In the cases of N-doped hydrochar, the activation process was carried out only with KOH under the same operating conditions. All activated materials were washed with 0.1 M HCl or NaOH and rinsed with deionized water to neutral pH. They were then recovered by filtration and finally dried at 105 °C overnight. The activated hydrochars were denoted, including the activating agent. For instance, HCN3-KOH represents the activated carbon obtained with KOH from the olive stone N-doped hydrochar with an initial mass ratio, C/N, of 3.

### 2.3. Characterization of Hydrochar and Activated Hydrochars

The proximate analyses were performed according to ASTM methods D3173-11 (moisture), D3174-11 (ash) and D3175-11 (volatile matter (VM)) using a Mettler Toledo apparatus (TGA/SDTA851e). The elemental analysis (C, N, S and H) was determined in a LECO Model CHNS-932 elemental analyzer. The analyses were performed in triplicate, with the standard deviation being less than 5% in all the cases.

$N_2$ adsorption-desorption isotherms at 77 K were performed in a Micromeritics apparatus (Tristar 3020). The samples were previously outgassed at 150 °C and a residual pressure of 10$^{-3}$ Torr for 6 h in a Micromeritics VacPrep 061 device. The surface area was calculated by the BET equation and the micropore volume ($V_{micro}$) was obtained by the t-method. The difference between the volume of $N_2$ adsorbed at 0.95 relative pressure (as liquid) and the micropore volume was taken as the mesopore volume ($V_{meso}$).

Scanning Electron Microscope (SEM) images of the hydrochar and activated hydrochar samples were obtained in Hitachi S-3000N apparatus. The samples were metalized with gold using a Sputter Coater SC502. Images were obtained in the high vacuum mode under an accelerating voltage of 20 kV, using secondary and back-scattered electrons.

The $pH_{slurry}$ of the activated hydrochars was determined by measuring the pH (pH-meter, Crison) of an aqueous suspension of the sample (1 g) in deionized water (10 mL) after being stirred overnight [52].

*2.4. Adsorption Tests*

The potential application of activated hydrochar ($\approx$100 μm particle size) as an aqueous phase adsorbent was assessed using sulfamethoxazole (SMX) as a model compound. SMX has a solubility in water of 610 mg L$^{-1}$ at 298 and a pKa of 1.7/5.6 [53]. Adsorption tests were performed by adding activated hydrochar (12.5 mg) into aqueous SMX solutions (25 to 300 mg L$^{-1}$) in 50 mL glass bottles. A commercial activated carbon (C: 89.5 wt.%., A$_{BET}$: 800 m$^2$ g$^{-1}$; V$_{micro}$: 0.67 cm$^3$ g$^{-1}$; V$_{meso}$: 0.53 cm$^3$ g$^{-1}$; pH$_{slurry}$: 7.7), supplied by Merck, was used as a reference for comparison purposes. Experiments were carried out in a thermostatized shaker bath (Optic Ivuymen System) at 20 °C, 200 rpm and the natural pH of the SMX solution (4.6) for 72 h, which was more than enough time to reach equilibrium. SMX concentration was determined by UV-vis spectrophotometry (Cary 60 UV-Vis Agilent Technologies) at 265 nm. The reported results are the average values from triplicate runs, being the standard errors below 5%. Langmuir and Freundlich equations were used to fit the equilibrium data and parameters were calculated using Origin 9.1 software.

## 3. Results

### 3.1. Characterization of the Hydrochars and Activated Hydrochars

Table 2 shows the proximate and ultimate analyses of the feedstock and hydrochars. Elemental analysis indicates that the raw material has the typical mass composition of an olive stone: carbon ($\approx$43–50 wt.%), oxygen ($\approx$43–49 wt.%), hydrogen ($\approx$6–7 wt.%) and low nitrogen and sulfur content ($\leq$0.4–0.1 wt.%) [54–58]. In the case of non-doped hydrochar, the carbonization process led to an increase in C content ($\approx$34 wt.%) and a decrease in oxygen ($\approx$36–42 wt.%) and hydrogen ($\approx$14–17 wt.%) content, because of the dehydration and decarboxylation reactions [59,60]. A reduced ash content (less than 0.8 wt.%) was observed in all cases. Regarding hydrochar yields, calculated as the mass of the hydrochar per unit mass of olive stone on a dry basis, values around 48–49 wt.% were obtained, regardless of the olive stone/water ratio (30–50 wt.% of OS on a dry basis) used in the HTC runs.

**Table 2.** Main characteristics of feedstock and hydrochars.

| Sample | Biomass/ Water (wt.%) | Proximate Analysis (wt.% Dry Basis) | | | Ultimate Analysis (wt.% Dry Basis) | | | | | A$_{BET}$ (m$^2$ g$^{-1}$) |
|---|---|---|---|---|---|---|---|---|---|---|
| | | Volatile Matter | Fixed Carbon | Ash | C | H | N | S | O * | |
| OS | - | 77.2 | 22.4 | 0.4 | 49.9 | 5.8 | 0.1 | 0.1 | 43.7 | - |
| HC | 30 | 52.5 | 47.1 | 0.4 | 66.7 | 4.8 | 0.2 | 0.0 | 27.9 | 18 |
| | 40 | 57.0 | 42.2 | 0.8 | 67.8 | 5.0 | 0.2 | 0.1 | 26.1 | 17 |
| | 50 | 56.1 | 43.3 | 0.6 | 67.8 | 4.9 | 0.2 | 0.0 | 26.5 | 15 |
| HCN2 | 50 | 64.7 | 35.0 | 0.3 | 54.0 | 4.8 | 7.7 | 6.4 | 26.8 | 5 |
| HCN3 | | 59.9 | 39.5 | 0.6 | 62.2 | 4.7 | 5.6 | 3.2 | 23.7 | 10 |
| HCN5 | | 60.5 | 39.1 | 0.4 | 63.9 | 4.9 | 4.9 | 1.4 | 24.5 | 4 |

* Calculated by difference O = 100 − (C + H + N + S + Ash).

N-doped hydrochars (HCN2, HCN3, HCN5) were prepared with an olive stones/water ratio of 50 wt.%, since it allows one to obtain a higher amount of hydrochar (Table 2). The C/N ratio in the initial solution (2, 3 and 5) was inferior to that of the hydrochar (7, 11 and 13) because of a less significant increase in carbon content than in nitrogen associated with the massive incorporation of nitrogen into the hydrochar structure, while no significant variations in the ash content were found. This fact has been observed previously in the literature using different nitrogen precursors (($NH_4)_2SO_4$, $C_5H_5NO$, $C_2H_5NO_2$) [16,18,20]. Moreover, the use of ($NH_4)_2SO_4$ as a doping agent also provoked an increase in sulfur content in these hydrochars.

The $N_2$ adsorption-desorption isotherms at 77 K of the hydrochars (Figure 1) are associated with solids with low porosity. They showed very low adsorption at low relative pressures revealing a limited porosity in all cases with specific surface areas values in the range of 4–18 $m^2 g^{-1}$. The N-doped materials showed the lowest surface area values, below 10 $m^2 g^{-1}$, but with a significant contribution of mesoporosity. Table 3 summarizes the porous structure of the activated hydrochars. Representative $N_2$ adsorption-desorption isotherms are depicted in Figure 1. All of them correspond to essentially microporous solids with a higher contribution of mesoporosity for the HC-$H_3PO_4$ and N-doped materials. For comparison, the olive stones were also chemically activated, without hydrothermal carbonization treatment (Figure 1). The porous structure developed by these materials was much lower than that the obtained from the hydrochars, achieving BET surface area values of 211, 504 and 254 $m^2 g^{-1}$ for olives stones activated with KOH, $H_3PO_4$ and $FeCl_3$, respectively, showing the crucial role of the HTC treatment in the development of the textural characteristics of activated hydrochars [44]. In the case of hydrochars, activation with $FeCl_3$ gave rise to a material with a relatively low surface area, slightly superior to the obtained by direct activation, while activation with $H_3PO_4$ resulted in a microporous material with a much higher relative contribution of mesoporosity, reaching BET surface area values of 1100 $m^2 g^{-1}$. Similar BET surface area values were obtained by direct activation of olive stones with steam, operating at 900 °C, a temperature significantly higher than those used in this work [61]. Activation with KOH allowed a high porosity development, essentially microporous, with BET surface area values up to around 2000 $m^2 g^{-1}$ and a significant ash content (17.1 wt.%). The activation treatments caused some important changes in the surface characteristics of the materials, specifically in the $pH_{slurry}$ value. As described by other authors, activation with $H_3PO_4$ resulted in a material with an acidic surface, while activation with KOH led to a material with a more basic surface [13,62]. Finally, activation with $FeCl_3$ gave rise to an activated hydrochar with a slightly acidic surface.

**Table 3.** Main characteristics of feedstock and hydrochars.

| Sample | C * (%) | N * (%) | Ash * (%) | $A_{BET}$ $(m^2 g^{-1})$ | $V_{micro}$ $(cm^3 g^{-1})$ | $V_{meso}$ $(cm^3 g^{-1})$ | $pH_{slurry}$ |
|---|---|---|---|---|---|---|---|
| HC-$FeCl_3$ | 47.0 | 0.3 | 10.0 | 383 | 0.18 | 0.07 | 6.5 |
| HC-$H_3PO_4$ | 70.2 | 0.2 | 9.1 | 1155 | 0.50 | 0.20 | 1.8 |
| HC-KOH | 74.4 | 0.1 | 17.2 | 2122 | 0.96 | 0.14 | 8.0 |
| HCN2-KOH | 49.6 | 0.99 | 31.7 | 1247 | 0.13 | 0.60 | 2.3 |
| HCN3-KOH | 40.2 | 0.63 | 29.1 | 1116 | 0.21 | 0.55 | 2.8 |
| HCN5-KOH | 59.2 | 1.52 | 25.1 | 2048 | 0.86 | 1.24 | 4.2 |

* C, N and Ash are represented as wt.% dry basis.

The potential of KOH as an activating agent was also tested in the activation of N-doped hydrochar, giving rise to materials with high surface area (1100–2000 $m^2 g^{-1}$), but significantly lower than that shown by the HC-KOH activated hydrochar, especially in the cases where the initial C/N mass ratio was lower (HCN2 and HCN3). This fact is related to the high ash content of these materials since the BET surface area represents up to about 1574–2734 $m^2 g^{-1}$ on an ash-free basis. Regarding pore size, the high N content in the precursor material favored the synthesis of mesoporous activated carbons [20]. The C/N ratio increased significantly after the activation treatment, indicating that the high temperature of the activation process caused the transformation of N into volatile N compounds leading to the loss of surface functional groups and resulting in the production of more stable N-containing species in the carbonaceous structure [63].

As previously discussed, the activation treatment causes some changes in the $pH_{slurry}$ of the activated hydrochars (Table 3). In the case of the N-doped hydrochars, the low values of $pH_{slurry}$ (2.3–4.8), despite being activated with KOH, could be attributed to the N-doping

agent ($(NH_4)_2SO_4$) causing a modification of oxygen groups leading to an increase in carboxylic acids [18].

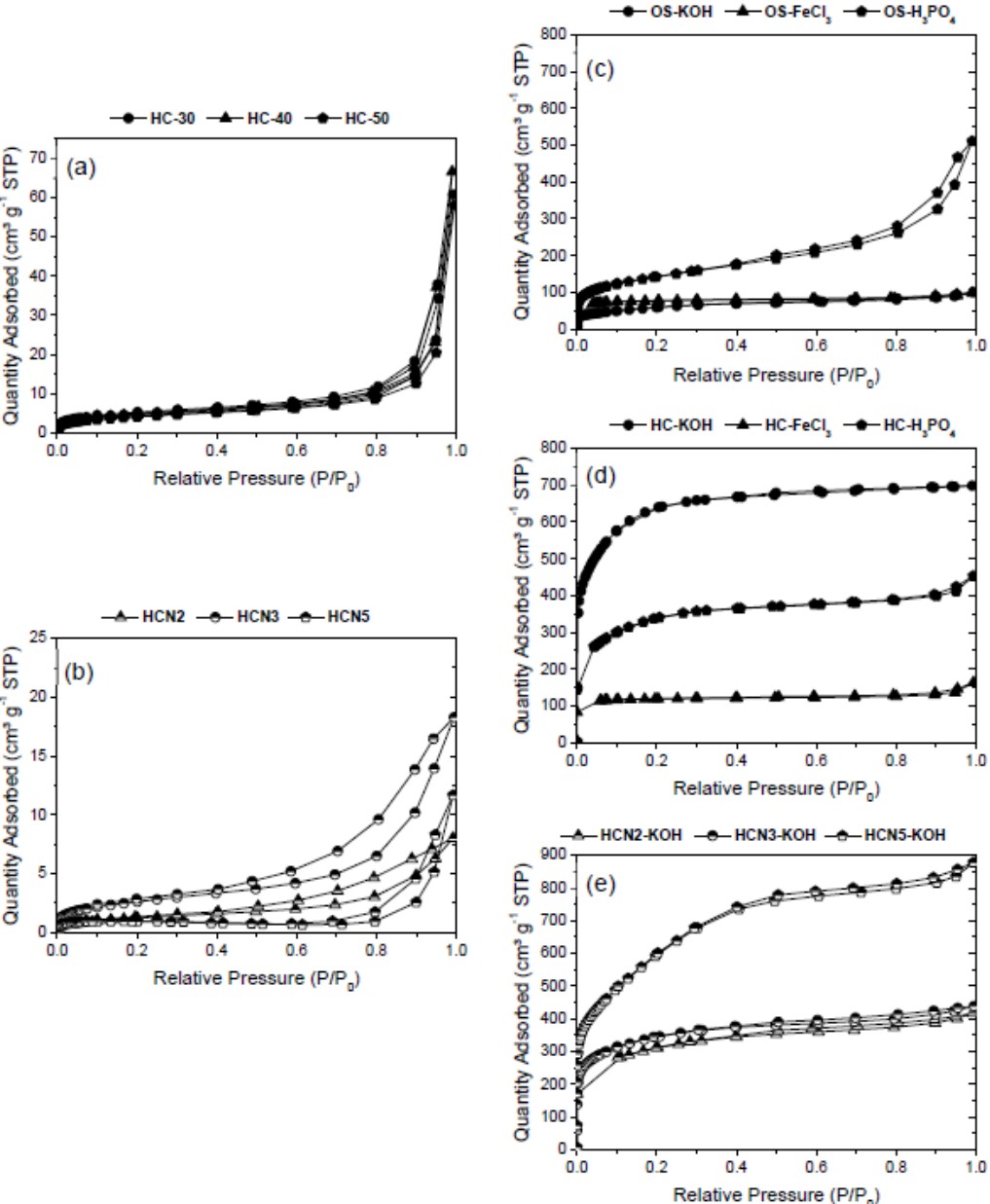

**Figure 1.** $N_2$ adsorption-desorption isotherms at 77 K of (**a**) hydrochar obtained at different biomass/water ratio (30–50 wt.%), (**b**) N-doped hydrochar, (**c**) materials obtained by direct activation of olive stones, (**d**) activated hydrochar and (**e**) activated N-doped hydrochar.

Figure 2 shows the scanning electron micrographs (SEM) of the surface of the olive stones, hydrochars and activated hydrochars. The raw olive stones presented a well-defined and regular structure, while the hydrochars showed some partial degradation of the surface, maintaining their original morphology, indicating the good thermo-mechanical stability of the raw material. The N-doped hydrochars showed multiple spheres of 1–5 μm diameter over the surface, originating in the doping process with $(NH_4)_2SO_4$. From the hydrochar images, non-porous materials were observed. However, activated hydrochars showed high porosity, with different structures depending on the activating agent. The hydrochars activated with KOH (HC-KOH and HCN5-KOH) showed cavities and cracks on their external surface, more regular in the case of the non-doped material. In the case of

FeCl$_3$-activation, the formation of microspheres was observed, while after activation with H$_3$PO$_4$, prismatic particles can also be seen.

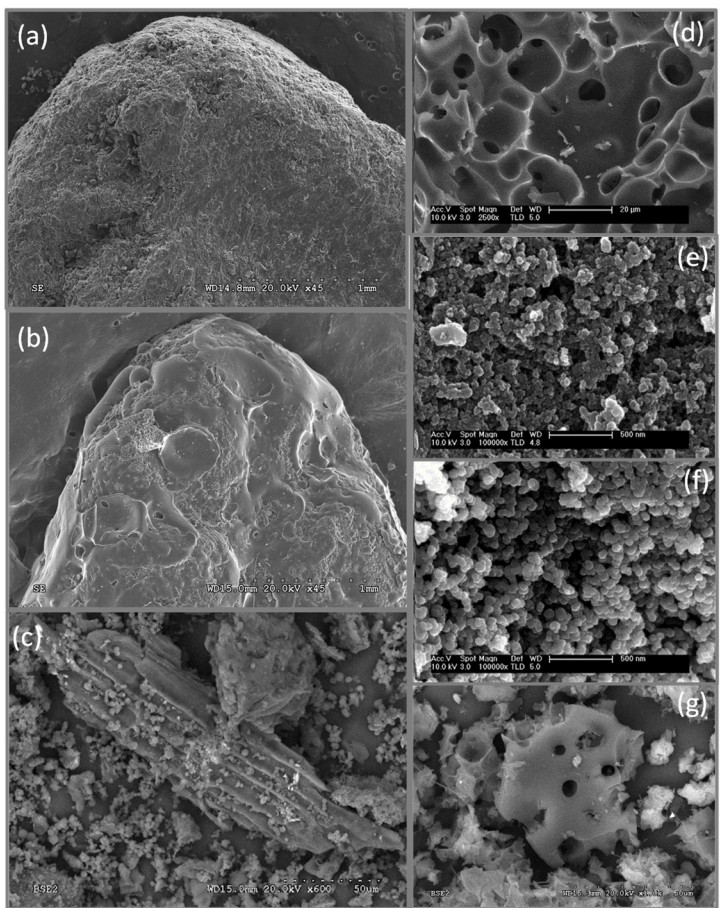

**Figure 2.** SEM images of (**a**) OS, (**b**) HC-50, (**c**) HCN5, (**d**) HC-KOH, (**e**) HC-FeCl$_3$, (**f**) HC-H$_3$PO$_4$ and (**g**) HCN5-KOH.

### 3.2. Adsorption of Sulfamethoxazole

The activated hydrochars were used in the SMX adsorption experiments, together with a commercial activated carbon, whose main characteristics were included in Section 2.4. for comparison purposes. Figure 3 shows the adsorption isotherms at 20 °C. The experimental data were fitted to the Langmuir (1) and Freundlich (2) equations:

$$q_e = \frac{q_L \cdot K_L \cdot C_e}{1 + K_L \cdot C_e} \tag{1}$$

$$q_e = K_F \cdot C_e^{\frac{1}{n}} \tag{2}$$

where $q_e$ is the equilibrium adsorbate loading onto the adsorbent (mg g$^{-1}$); $C_e$, the equilibrium liquid phase concentration of adsorbate (mg L$^{-1}$); $q_L$, the monolayer adsorption capacity of the material (mg g$^{-1}$); $K_L$, the Langmuir constant (L mg$^{-1}$); $K_F$, the Freundlich constant ((mg g$^{-1}$)·(L mg$^{-1}$)$^{(1/n)}$) and n is the adsorption intensity. The values of the fitting parameters and correlation coefficients are given in Table 4. As can be seen, the results fitted the Freundlich equation better than the Langmuir equation, especially in the case of the adsorbents from HC. The HC-KOH material showed the highest adsorption capacity of the Langmuir monolayer, reaching almost the value of 760 mg g$^{-1}$, followed by the materials obtained by KOH activation of N-doped hydrochars ($q_L$: 429–695 mg g$^{-1}$). These adsorption capacity values are comparable to or even superior to those obtained with

adsorbents prepared from other biomass wastes (Table 1). Regarding Freundlich isotherms, the value of the "$n$" was lower than 1 (0.17–0.33) associated with favorable isotherms, with a high amount of adsorbate adsorbed at low concentrations [64]. Except in the case of the HC-$H_3PO_4$ material, values of "$n$" were between 0.17 and 0.22, so an approximation of the adsorption capacity of the materials could be established from "$K_F$" values with the following sequence: HC-KOH > HCN5-KOH > HCN2-KOH > HCN3-KOH >> commercial AC > HC-FeCl$_3$. Attending to the value of "$q_L$", calculated from the Langmuir equation, the resultant sequence would be the same, placing HC-$H_3PO_4$ between HCN3-KOH and commercial AC.

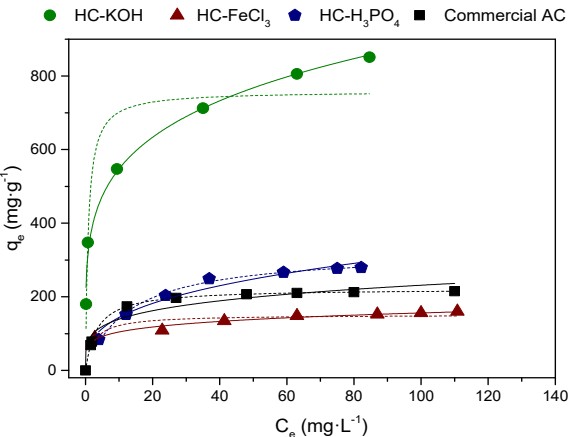
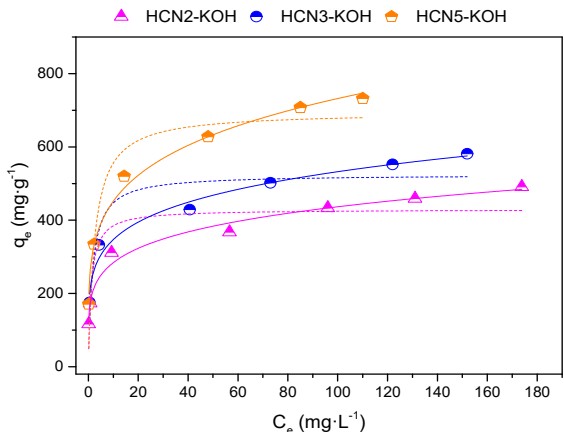

**Figure 3.** Adsorption isotherms of SMX on HC-KOH, HC-FeCl$_3$, HC-$H_3PO_4$, HCN2-KOH, HCN3-KOH and HCN5-KOH and commercial activated carbon at 20 °C (symbols: experimental values; short dot lines: fitting to the Langmuir equation; solid lines: fitting to the Freundlich equation).

**Table 4.** Langmuir and Freundlich parameters for SMX adsorption on the activated carbons of Figure 3.

| Sample | Langmuir | | | Freundlich | | |
|---|---|---|---|---|---|---|
| | $q_L$ (mg g$^{-1}$) | $K_L$ (L mg$^{-1}$) | $R^2$ | $K_F$ (mg g$^{-1}$)·(L mg$^{-1}$)$^{(1/n)}$ | n | $R^2$ |
| Commercial AC | 221.8 ± 9.2 | 0.296 ± 0.071 | 0.951 | 83.8 ± 12.1 | 0.220 ± 0.036 | 0.938 |
| HC-FeCl3 | 151.2 ± 7.6 | 0.419 ± 0.187 | 0.673 | 69.9 ± 5.7 | 0.174 ± 0.019 | 0.945 |
| HC-H3PO4 | 324.4 ± 7.4 | 0.078 ± 0.007 | 0.992 | 66.3 ± 10.4 | 0.337 ± 0.039 | 0.947 |
| HC-KOH | 758.7 ± 55.9 | 1.236 ± 0.630 | 0.855 | 338.3 ± 18.3 | 0.209 ± 0.014 | 0.987 |
| HCN2-KOH | 524.6 ± 34.2 | 0.579 ± 0.290 | 0.830 | 226.2 ± 16.8 | 0.186 ± 0.017 | 0.975 |
| HCN3-KOH | 429.4 ± 29.3 | 0.862 ± 0.464 | 0.829 | 184.8 ± 11.5 | 0.187 ± 0.014 | 0.982 |
| HCN5-KOH | 695.4 ± 48.9 | 0.397 ± 0.185 | 0.878 | 278.9 ± 14.6 | 0.209 ± 0.013 | 0.989 |

The adsorption capacity of the activated hydrochars was mainly influenced by the BET surface area of the carbons followed by the existence of acid or basic surface functional groups. This fact was also observed in the adsorption of methylene blue using KOH-modified hydrochar from sewage sludge, where the adsorption process was the result of several phenomena such as physi- and chemisorption, acid-base and redox equilibria [65]. In this study, consistently with their much larger surface area, HC-KOH and HCN5-KOH yielded the highest adsorption capacity. Comparing these two materials, electrostatic interactions could explain the best performance of the HC-KOH material. At solution pH (4.6), SMX should be mostly a neutral and anionic species. Since the pH$_{slurry}$ of HCN5-KOH is 4.2, its surface will be negatively charged, while the surface of the HC-KOH

material will be positively charged, which will favor the interaction of the latter with the adsorbent [45,66,67]. Then, the effect of N doping did not have a positive effect on the adsorption of the SMX to provide positively charged functional groups on the surface of the adsorbent. A comparison of the adsorption capacity of HC-H$_3$PO$_4$, HCN2-KOH, HCN3-KOH materials, which are characterized by similar values of specific surface area (A$_{BET}$: 1115–1247 m$^2$ g$^{-1}$) and pH$_{slurry}$ lower than that of the solution, allows the evaluation of the effect of acidic or basic functional groups on the surface of the activated hydrochars on the adsorption capacity of SMX. The better performance of the N-doped materials, with similar properties to the HC-H$_3$PO$_4$ activated hydrochar could be attributed to the higher negative charge of the latter, whose pH$_{slurry}$ is like pKa 1 of SMX. Finally, the substantial differences in adsorption capacity between the HC-FeCl$_3$ and HC-H$_3$PO$_4$ materials, and the commercial AC, used as reference material, could be due to the lower surface area of the former since all of them present similar electrostatic interactions with the adsorbate.

## 4. Conclusions

Hydrothermal carbonization followed by chemical activation with KOH of olive stones can be a promising way of valorization of such biomass to produce activated hydrochars. Activation with FeCl$_3$, H$_3$PO$_4$ and KOH of hydrochar resulted in high BET surface area carbons with predominantly microporous structure and a neutral, acidic and basic surface, while KOH activation of N-doped hydrochar also provided activated carbons with a high BET surface area but with a significant contribution of mesoporosity and an acidic surface. These activated hydrochars can be used as adsorbents for pollutants in water, as can be deduced from the high adsorption capacity shown for sulfamethoxazole, based on the high BET area value and the electrostatic interaction between the adsorbent and the adsorbate.

**Author Contributions:** Conceptualization, E.D., C.J.C., A.F.M.; Methodology, E.D., I.S.; Investigation, E.D., I.S.; Writing—Original Draft Preparation, E.D., I.S., A.F.M.; Writing—Review and Editing, E.D., C.J.C., A.F.M.; Supervision, E.D., C.J.C., A.F.M.; Funding Acquisition, E.D., C.J.C., A.F.M. All authors have read and agreed to the published version of the manuscript.

**Funding:** The authors greatly appreciate the financial support from the Spanish MICIIN (PID 2019-108445RB-100), Comunidad de Madrid (S2018/EMT-4344), US National Science Foundation (NSF #CBET-1856009), and UAM-Santander (2017/EEUU/07). I. Sanchis wishes to thank the Comunidad de Madrid for PEJD-2017-PRE/AMB-4616 contract.

**Institutional Review Board Statement:** Not applicable.

**Informed Consent Statement:** Not applicable.

**Data Availability Statement:** Not applicable.

**Acknowledgments:** The authors thank F.J. Manzano for his valuable help.

**Conflicts of Interest:** The authors declare no conflict of interest.

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
