# Peer review of "Activated Carbons from Hydrothermal Carbonization and Chemical Activation of Olive Stones: Application in Sulfamethoxazole Adsorption"

_resources, doi:10.3390/resources11050043_

Round 1

Reviewer 1 Report

The paper “Activated carbons from hydrothermal carbonization and chemical activation of olive stones. Application in sulfamethoxazole adsorption” is generally well conceived and structured with interesting data which are properly analyzed and discussed. The use of the English language is quite good, even if it needs improvement in some points (some of which are listed below). The aspects to improve in the review of the article are listed below, point by point.

Line 88 <<than those other activated carbons prepared from algae>>: than those of other…

Line 96 <<to prepare activated carbons to amoxicillin adsorption>>: … carbons for amoxicillin adsorption…

Table 1, caption. <<from water by biomass adsorbents>>. I think much better is: biomass-derived (or based) adsorbents.

Line 129 <<using 20 g of dried olive stones and various water to OS ratios (30-50 wt.% of OS). The reactor was inserted in a muffle furnace (Hobersal serie 8B Mod 12 PR/400) and heated up to 220 °C for 16 h [45].>>. It is not so clear the meaning of OS ratio: is it biomass to water ratio? Moreover, the OS ratio range foresees very high (up 50 wt.%) values. I suggest that the authors clarify about the definition of OS ratio and comment-motivate why they have operated at such high values of the ratio. Even more important is to explain the reason for a reaction time of 16 hours, which is well beyond the typical reaction times of the HTC process, both those in the scientific literature and those used industrially.

Line 178 <<The potential application of activated hydrochar (≈ 100 micro m) as an aqueous phase>>. What does the numerical value in brackets mean? It needs to be clarified (or removed).

Line 199 (and Table 2) <<A reduced ash content (less than 1.8 %) was observed in all cases.>>. The ash value (1.8%) of the HC sample obtained at 40% OS ratio is out of trend and cannot be considered reliable and physically consistent. Given the hydrochar yield (48-49%) and the ash content of the OS feedstock (0.4%), the value of 1.8% cannot be considered consistent. I suggest the authors to repeat the proximate analysis of that sample to provide a reliable value of the ash content of such a HC sample.

Line 204 <<was inferior to that the hydrochar>>: that of the hydrochar.

Line 254 <<precursor material allowed obtained mesoporous activated carbons>>. Revise the English.

Table 3. The authors should try to compare the values they obtained with others available in the literature. For instance, the pH value of the HC-KOH sample (8.0) agrees with the pH values measured in KOH activated HC samples obtained from grape seeds: Purnomo et al., Granular activated carbon from grape seeds hydrothermal char, Applied Sciences 8 (2018) 331. http://doi.org/10.3390/app8030331

Line 296 <<Figure 2 shows the scanning electron micrographs (SEM) of the surface morphology of the olive stones… >>. Delete morphology.

Line 337 <<especially in the case of the home-made adsorbents>>. Rephrase: I think home-made is not appropriate.

Line 350 <<The adsorption capacity of the activated hydrochars was mainly influenced by the BET surface area of the carbons followed by the existence of acid or basic surface functional groups.>>. This, and the following discussion <<… electrostatic interactions could explain the best performance of the HC-KOH material.>> can benefit from the in deep analysis performed on KOH modified HCs prepared starting from sewage sludge: Ferrentino et al., Sewage sludge hydrochar: an option for removal of methylene blue from wastewater, Applied Science 10 (2020) 3445 https://www.mdpi.com/2076-3417/10/10/3445

Overall I confirm my very positive opinion of the manuscript: after improving the parts I highlighted above, the paper can be accepted for publication.

Reviewer 2 Report

This is a well-written article regarding the production of activated carbons through HTC and the chemical activation of olive stones. Authors focused on comparing different activation conditions on the characterization and implementation potential of resultant bioproducts (here, adsorption of SMX as an exemplary hazardous compound). The reviewed work is concise and authors clearly presented motivation, methodology and results. Several interesting conclusions were obtained. As such, this work is recommended for publication.
Minor comments which author could take into account: (i) Authors could specify the type of % through the manuscript – direct (% wt.) or relative (sometimes it is missing e.g. in Section 3.1) (ii) Table 2 – proximate analysis – the sum is around 90-95 wt.% - what is else? (iii) please comment why for HC_40 the ash content is 3-4 times higher than for other samples? (iv) please enhance the quality of  Fig. 3. 
